# The Impact of Sex on the Response to Proton Pump Inhibitor Treatment

**DOI:** 10.3390/ph16121722

**Published:** 2023-12-12

**Authors:** Holmfridur Helgadottir, Einar S. Björnsson

**Affiliations:** 1Department of Internal Medicine, Haraldsplass Diaconal Hospital, 5009 Bergen, Norway; hofihelgad@gmail.com; 2Department of Gastroenterology, Haukeland University Hospital, 5009 Bergen, Norway; 3Department of Gastroenterology, Landspitali University Hospital, 101 Reykjavik, Iceland; 4Faculty of Medicine, University of Iceland, 101 Reykjavik, Iceland

**Keywords:** proton pump inhibitors, acid suppression, gastrin, pharmacokinetics, biological sex

## Abstract

Proton pump inhibitor (PPI) treatment is responsible for substantial gastrin elevation secondary to reduced intragastric acidity. Due to the increasing global prevalence of PPI users, concerns have been raised about the clinical significance of continuous gastrin elevation and its potential long-term side effects. Hypergastrinemia secondary to PPIs has trophic effects on gastric mucosa, leading to enterochromaffin-like cell hyperplasia and gastric (fundic) polyp formation, and it is believed to provoke acid rebound following PPI withdrawal that induces PPI overutilization. Previous studies have found higher gastrin release following PPI therapy in females compared with males, and sex differences have also been demonstrated in pharmacokinetic parameters and dose requirements for acid reflux. It is conceivable that females might be at increased risk of PPI overuse, because they often receive higher milligram-per-kilogram doses. The prevalence of PPI use is more common among females, and the female sex is a risk factor for adverse drug reactions. This non-systematic review outlines the current knowledge of the impact of biological sex on the response to PPIs. The aim is to highlight the female sex as a potential risk factor that could be a step toward precision medicine and should be considered in future research on the response to PPI treatment.

## 1. Introduction

A recent literature review showed that compared with males, females were involved in more adverse drug reaction (ADR) reports [1]. Studies on specific types of drugs illustrating this sex difference and the underlying pathophysiology are important. Proton pump inhibitors (PPIs) are by far the most used acid-suppressive drugs and constitute the first-choice treatment for gastric acid-related disorders [2]. The major adequate indications for their use are treatment of gastro-esophageal reflux disease, peptic ulcers, eradication of *Helicobacter pylori* infection in combination with antibiotics, and ulcer prophylaxis for patients on non-steroidal anti-inflammatory or antiplatelet drugs [3]. Since their introduction in clinical practice over 30 years ago, their therapeutic success has been accompanied by a dramatic rise in the prevalence of PPI users [4]. PPIs are currently among the most prescribed of all drugs, and the rate of PPI use continues to increase. Similar to other drugs, PPIs have some side effects, and there is considerable concern regarding their long-term safety. One of these long-standing concerns is PPI-induced gastrin elevation secondary to hypoacidity. Profound acid inhibition by PPIs leads to gastrin elevation. Gastrin is a growth hormone, and concerns have been raised regarding its potent trophic effects: gastric polyp formation and potential progression to dysplasia following long-term acid suppression [5]. PPI-induced gastrin elevation is also believed to play a role in rebound hyperacidity when PPIs are discontinued [6,7,8,9]. This could induce dyspeptic symptoms that might result in the resumption of PPI therapy. This rebound acid hypersecretion (RAHS) is believed to result from gastrin’s hypertrophic effects on enterochromaffin-like (ECL) cells in the stomach [8,9]. RAHS leads to increased acid production after therapy is discontinued, which might partly explain the increasing prevalence of overall PPI use [6,7]. PPIs are generally well tolerated and considered safe drugs, which might also play a role in the overutilization and inappropriate long-term use of the drugs [10,11]. Therefore, using the lowest effective dose for the shortest possible time has been the mainstream method employed to promote long-term safety. Careful consideration by prescribers of the appropriate indication, as well as the dose and duration of treatment, is necessary in the determination of the benefit–harm balance of PPI therapy. Only a few studies have aimed to investigate the impact of sex on PPI treatment, and some of the findings on sex differences indicate that compared with males, females might be more sensitive to PPIs’ inhibitory effects on acid secretion and induced gastrin release [12,13]. Gender difference may also exist in PPI indications, symptom perception, and other pre-treatment factors not related to acid regurgitation that affect the PPI response [14,15,16]. One of these factors is PPI metabolism and pharmacogenetics [17]. A review of 12 pharmacokinetic studies from 2001 suggested a sex-related difference in the pharmacokinetic parameters, where the area under the curve (AUC) and maximum concentrations (C_max_) values were approximately 30% higher in females than in males after a single dose [18]. The practitioner’s consideration of sex and gender as a possible factor contributing to PPI treatment response is not a mainstream approach but might be appropriate. This review is based on a non-systematic search in PubMed and the authors’ own clinical experience and research, and the aim is to highlight a few reasons why considering the female sex as a risk factor could be an appropriate step toward precision medicine that would probably benefit individualized treatment. In this review, the term sex refers to the biological and physiological characteristics, and gender refers to the socioeconomic or cultural distinctions and roles associated with being male or female [19].

## 2. Association between Female Sex and Increased Gastrin Release Following PPIs

It is widely accepted that all individuals undergoing PPI treatment develop varying degrees of gastrin elevation [20,21]. Currently, there is limited information on which individuals are likely to develop significant hypergastrinemia, defined as gastrin levels above the upper limit of the reference range for fasting blood gastrin. Serum gastrin levels during PPI treatment have been positively correlated with basal gastrin levels prior to PPI treatment [22,23]. This suggests an individual effect, beyond PPI and acid inhibition itself, that likely contributes to enhanced gastrin levels. Demographics that have been associated with higher gastrin levels include *H. pylori* infection [24], atrophic gastritis [25], age [26], and sex [20,21]. Shortly after the introduction of PPIs in the late 1980s, it was reported that females were more likely to develop higher gastrin levels early in the course of omeprazole treatment, reaching statistically significant differences at 18 and 21 months (Figure 1) [22]. Since the turn of the 21st century, many researchers have studied PPIs’ long-term effect on gastrin, mainly on the fasting serum gastrin values, but only a limited number of these studies have included sex-disaggregated data [27,28]. A significant sex difference was reported again in 2014, with higher basal and meal-stimulated serum gastrin levels among females than males on long-term PPI therapy for gastroesophageal reflux disease (GERD), whereas such sex difference was not observed in healthy controls who were not on PPIs (Figure 2) [28].

Previous studies that examined sex difference in gastrin response among healthy subjects and patients showed conflicting results, while suggesting a sex-related difference in gastrin release. The results of 11 studies with published sex-disaggregated data on gastrin release and/or gastrin elevation in healthy volunteers and patients on PPIs are listed in Table 1. Interestingly, if sex difference was found, gastrin levels were higher in females than in males. Despite the sex difference among healthy subjects, as reported in the literature throughout the 20th century, sex difference is rarely documented in papers about PPI-induced gastrin elevation. Sex difference in gastrin release among PPI users is still incompletely defined, with a limited number of studies examining sex-related differences. Likewise, most of the studies listed in Table 1 measured fasting serum gastrin levels, whereas only a few studies evaluated meal-stimulated gastrin concentrations and the area under the meal-stimulated gastrin curve, which are better indicators of hypergastrinemia [30,31]. A single fasting gastrin measurement might underestimate the hormone production, because gastrin levels are at their lowest point during overnight fasting. Daily fluctuations in gastrin levels continue in patients on PPIs—but to a higher degree [32]. High degrees of intra- and inter-individual variations in serum gastrin levels were observed in previous studies, supporting the argument that repeated sampling of gastrin levels on different days or meal-stimulated measurements might provide better estimates of chronic gastrin elevation [27,28,31]. The reason for the observed sex difference in gastrin response induced by PPI therapy remains unclear. Table 2 presents a list of possible factors that could contribute to sex-related differences in gastrin release in healthy subjects and gastrin elevation secondary to PPI therapy.

## 3. Role of Sex in Metabolism of PPIs

PPIs undergo low rates of first-pass hepatic metabolism, and the oral bioavailability of PPIs is high. Additionally, the rate of protein binding is >95% and independent of sex [18,47]. PPIs are metabolized in the liver by cytochrome P450 (CYP) enzymes, known to show marked inter-individual and inter-ethnic variations. CYP2C19 and CYP3A4 are the two enzymes mainly involved in the metabolism of PPIs [18,45]. All PPIs, except rabeprazole, rely significantly on CYP2C19 for their clearance [48]. Sex difference in the expression and activity of CYP2C19 and CYP3A4 has been reported [46,49,50,51,52]. Generally, PPIs have similar efficacy, although they have some differences regarding their pharmacokinetics and pharmacodynamics. PPI effectiveness can also vary, depending on each PPI’s affinity to the CYP2C19 enzyme.

A review of 12 pharmacokinetic studies suggested a sex difference in two pharmacokinetic parameters [18]. The area under the curve (AUC) and maximum concentration (C_max_) values were approximately 30% higher in females than in males after a single dose, with less difference during repeated administration of esomeprazole 40 mg during fasting conditions [18]. This might be due to the differences in CYP2C19 and CYP3A4 expression between the two sexes. It was previously shown that females had higher CYP3A4 activity than males, whereas CYP2C19 activity was lower in females [50]. In a large population study involving Dutch Caucasians, CYP2C19 activity was 40% greater in males than in females [51]. Other possible explanations might be differences in body weight (higher in males than in females) or volume of distribution (V_d_), with females having a higher percentage of body fat than males [46]. A study in Iran found that females on omeprazole had higher weight-normalized V_d_ and clearance (Cl) [45]. The authors suggested that these sex differences might be explained by sex differences in metabolism and body fat percentage, respectively [45]. However, another study in Iceland on the pharmacokinetics of esomeprazole in healthy males and females found no sex difference in the pharmacokinetic parameters after a single dose or repeated doses for five days [38].

The pharmacokinetic parameters may also greatly differ by the CYP2C19 genotype, including extensive metabolizers (EMs), intermediate metabolizers (IMs), and poor metabolizers (PMs), and this can cause considerable inter-individual and -ethnic variation in the PPI metabolism [17]. PPIs are rapidly eliminated from systemic circulation in individuals with the EM genotype, resulting in markedly lower plasma PPI concentrations than those observed in individuals with the PM genotype [47]. The genotype may therefore influence the PPI response, with risk of under-treatment (partial symptom response or non-response) among EMs and risk of over-treatment among PMs (adverse events, unnecessary expense) [17]. In one of the studies that reported hypergastrinemia more frequently in females on long-term PPI use, CYP2C19 polymorphisms were also assessed, but no significant difference in median gastrin levels was found dependent on CYP2C19 gene polymorphisms (EM vs. IM vs. PM) [20]. It remains unclear whether the pharmacokinetics of PPIs play a role in the previously described sex difference in secondary gastrin response (other possible factors are listed in Table 2). 

## 4. Is There a Need for Sex-Specific PPI Dosage?

The female sex and PPI dosage seem to perform an important function in the development of hypergastrinemia during PPI treatment [21]. Several studies have found higher gastrin levels in patients on higher PPI doses [21,53,54,55]. The elevated serum gastrin levels observed in patients exposed to higher PPI doses (in milligrams per day) are likely the effect of gastric acid inhibition. Gastrin elevation is correlated with the degree of acid inhibition by PPIs, measured by the fasting intragastric pH level [56,57]. It is known that females have smaller stomachs and parietal cell masses than males [39,40], and it is conceivable that the PPI doses used in females are higher than necessary to inhibit acid secretion in order to obtain symptomatic relief. In other words, compared with males, females might be more sensitive to PPIs’ inhibitory effects on acid secretion and induced gastrin release. A randomized clinical trial found that female gastroesophageal reflux disease (GERD) patients on long-term PPIs were three times more likely than males to tolerate a 50% reduction in their prior doses over an 8-week period [12]. The female sex was the only independent predictor of a successful step-down, though with only a modest odds ratio (OR: 1.3 [95% CI: 1.01–1.6]). In another step-down study, male patients were less likely to be candidates for a step-down because of the worse control of their GERD symptoms despite twice-daily PPI use initially [13]. This sex difference also did not reach statistical significance; female patients who did not retain heartburn control after the therapy switch had an OR of 0.499 (95% CI: 0.178–1.396) [13]. The reason why the sex difference did not reach statistical significance in these two studies might be the relatively small sample sizes of the step-down groups, with 50 (25 females) and 142 (80 females) participants, respectively [12,13]. To date, a limited number of deprescribing studies have provided information on the differences between the sexes (Table 3) [12,13,58,59]. Female participants were also underrepresented in previous step-down research, accounting for <5% of the total study population in a couple of studies [60,61]. However, the above-mentioned sex-specific findings on PPI dose requirements suggest that females need lower PPI doses than males do to obtain relief from acid-related symptoms. 

## 5. Overuse or Overmedication of PPIs among Females

In recent years, concerns have been raised about the increasing prevalence of patients on long-term PPI therapy. Although the incidence of new therapy with PPIs remains stable, the prevalence of PPI therapy continues to rise [4,62,63]. Some studies have shown that females generally use PPIs more often than males [4,62,64,65]. In a recent systematic review that described global PPI use patterns by demographics, females comprised 56% of PPI users in the general population; 60 of the 65 studies included in the analysis provided sex information [66]. The same review reported that nearly two-thirds of the users were on high doses (≥defined daily dose (DDD)) [66]. Similar to almost all types of drugs, PPI dosage has not been sex-specific, but females often receive higher milligram/kilogram doses than males. A nationwide drug utilization study from Iceland that described the outpatient PPI use among the entire adult population found that in addition to the rising prevalence of PPI use across time, the prevalence increased with higher age and was higher among females than males in all age groups (Figure 3) [62]. In addition, long-term PPI use (over 1 year) was highest among the elderly population, and they are also generally at greater risk for polypharmacy and adverse drug reactions because of the metabolic changes associated with ageing [62,67].

Serum gastrin was found to be an independent predictor of PPI requirement in a study on the discontinuation of PPIs after long-term treatment [58]. Studies on RAHS in patients after their long-term PPI therapy are largely lacking. RAHS is believed to contribute to difficulties in the discontinuation of treatment and in acid rebound, which might explain—at least partly—the increase in long-term users without an adequate indication of needing PPIs [68]. Concerns have been raised about this physical dependence on PPIs and its potential complications and economic consequences [69]. It has also been hypothesized that the increase in reflux disease incidence over recent decades may be due to the worsening of reflux symptoms caused by RAHS; in other words, PPI therapy for reflux symptoms might be worsening the disease itself [9]. It is conceivable that although females might need lower doses of PPIs to control their acid-related symptoms, they might be more at risk of RAHS due to elevated gastrin levels, which in turn might contribute to their higher prevalence of PPI use. However, this needs to be investigated in clinical studies.

In post-hoc analysis, partial PPI treatment response is more common in non-erosive reflux disease (NERD) than in reflux esophagitis [70] and has been associated with the female gender [14]. Since NERD is more common in females, this might explain the gender difference in PPI response, because NERD is less often caused by acidic reflux [16,70]. Among 580 GERD patients who responded partially to PPIs, a history of reflux esophagitis was significantly higher in males than in females (24% vs. 11%) [14]. In the same analysis, females had significantly more burdens from extra-esophageal symptoms, abdominal pain, indigestion, and constipation [14]. The prevalence of anxiety, depression, and use of antidepressants was also significantly higher among females than males [14]. The authors suggested that these comorbidities increased symptom burden in females and might contribute to their partial PPI response [14]. Another post-hoc analysis showed that females with GERD were more likely to need dose escalation because of inadequate symptom control [71]. Of the 99 patients (33 females) who needed dose escalation, one-third could return to the lower dosage over the coming years, but those patients’ characteristics were not mentioned [71]. In a multicenter prospective observational study, which included 182 Japanese patients with symptomatic GERD (62 females (34%)), a multiple regression analysis found the following pre-treatment factors to be associated with residual symptom rate: milder GERD symptoms, absence of esophagitis, severer epigastric pain/ burning symptoms, lower BMI, and severer depression [15].

PPIs are widely prescribed today, not only for acid-related disorders but frequently also for a variety of upper gastrointestinal (GI) conditions not necessarily related to acid, partly due to a lack of other therapeutic modalities for upper GI symptoms, often minor symptoms with unidentifiable causes (dyspepsia). PPIs are beneficial to those who have appropriate indications but obviously not to those who have inappropriate indications. With the increase in long-term PPI use in patients with inappropriate indications, the risks of long-term PPI use must be weighed against the benefits. Inappropriate PPI use has not been associated with any gender, but it has been associated with older age, non-steroidal anti-inflammatory and anti-platelet drugs [72], and polypharmacy in elderly patients [73].

As pointed out, the prevalence of global PPI use in the general population has been shown to be more common among females [66]. Among others, heartburn, regurgitation, and extra-esophageal symptoms have been more frequently reported by women than by men, suggesting that sex and gender play a role in symptom perception [16]. Research on lower PPI response among females also suggests that comorbid anxiety and depression may contribute to the increased symptom burden in females [14]. PPIs are also widely used by pregnant and breastfeeding women, since GERD is one of the most common medical complaints during pregnancy [74]. In contrast, males seem to have a higher prevalence of pathologic acid-related diseases, such as reflux esophagitis, Barrett’s esophagus (BE), and esophageal adenocarcinoma (EAC), that require continuous maintenance therapy with PPIs [16]. Differential sensitivity with augmented symptoms in females might have diagnostic and therapeutic influences. Therefore, it is important to consider other comorbidities and symptoms not associated with acid that may contribute to the increased symptom burden in females before attempting dose escalation in PPI partial responders or non-responders (Table 4).

## 6. Side Effects of Secondary Gastrin Elevation

Although PPIs are generally safe, their extensive and increasingly long-term use has led to concerns about the safety of long-term PPI treatment. Although the clinical significance of elevated fasting gastrin levels has not been fully determined, they have a well-documented association with both ECL hyperplasia and increased risk of gastric polyps in humans, especially fundic gland polyps (FGPs) [5,75]. FGPs are the most common gastric polyps arising within fundic gland mucosa in the fundus and body of the stomach [76]. They are known to be associated with PPI use and are generally considered benign [76]. A recent Chinese study on the characteristics of 186 gastric polyps found significantly more polyps in females than in males (with a ratio of 2.4:1), of whom 78% of the total study population were long-term PPI users (>5 years) [77]. In a population of 1005 patients, of whom 441 received acid-suppressive therapy, a multivariate analysis found the female sex to be associated with the prevalence of FGPs and multiple white elevated lesions in gastric mucosa [78]. Although gastrin was not measured in that study, the authors suggested that higher gastrin levels among females might influence the prevalence of multiple white elevated lesions, as well as FGPs [78]. Despite the low rate of dysplasia or carcinoma in FGPs, a summary of 11 cases found that the majority of them involved middle-aged women (65.7%), most of whom were receiving PPI therapy [79]. A theoretical link between gastrin elevation and gastric cancer was previously proposed [80]. Several large cohort studies suggest that PPI use is associated with an increased risk of gastric intestinal metaplasia and gastric cancer with duration and dose-dependent association [81,82]. However, most of these are retrospective observational studies hampered by important methodological weaknesses, and causal relationship has been difficult to demonstrate [81,82]. Although the reported risk is only possible, and the precise underlying mechanism is unclear, the increasing prevalence of PPIs worldwide remains a matter of concern.

## 7. Other Side Effects of PPIs

Gastrin elevation is only one of many concerns that have been raised parallel to the increase in long-term PPI use with or without appropriate indications [66,83]. The putative adverse effects of long-term PPI therapy are mainly obtained from observational studies in various patient populations with many unmeasured variables [84]. Some have been related to the PPI-induced hypochlorhydria, which has been associated with increased risk for infections (*Clostridium difficile* infection and pneumonia) and malabsorption of micronutrients (magnesium, calcium, and vitamin B12 deficiencies) [83,84]. Others are idiosyncratic, and the precise molecular mechanism of how they occur is often unclear. These include, for example, increased risk of bone fractures, kidney disease, cardiac disease, and dementia [83,84]. These associations, just like the above-mentioned cancer risk, should be interpreted with caution owing to the lack of high-quality studies exploring these relationships. Pharmacokinetic (PK) interactions are also important when PPIs are prescribed with other drugs, since PPIs competitively inhibit certain CYP enzymes (e.g., CYP2C19) to varying degrees [84].

A recent literature review showed that compared with males, females were involved in more ADR reports [1]. This might be due to gender-related factors, such as females’ higher tendency to report their health conditions compared with that of males. A cross-sectional study of 66,102 subjects who reported ADRs (while taking PPIs) to the Food and Drug Administration (FDA) database between 1997 and 2012 found that more females indicated one or more ADR (57.3%) [85]. Several gender differences in the adverse reactions to PPIs have been reported. A recent study in China demonstrated that women were more likely to develop the PPI-related syndrome of inappropriate antidiuretic secretion (SIADH) [86]. However, the risk of PPI-associated hypomagnesemia was higher in males and in the elderly population [85].

Many of the side effects associated with long-term PPI therapy have been shown to be dose-related [87]. In a recent placebo-controlled study, women on PPI therapy were three times more likely than men on PPI therapy to decrease their PPI doses by half [12]. If the majority of female patients on PPI therapy tend to be overmedicated and could reduce their PPI use, this could lead to major economic benefits for the healthcare system and potentially decrease the risk of side effects associated with long-term PPI use. As pointed out, the female sex has been shown to be a risk factor for clinically relevant ADRs [1,46].

## 8. Limitations

Our review includes some limitations that need to be addressed. First, this is a non-systematic review. Second, our hypothesis that females might be more sensitive to the inhibitory effect of PPIs is derived from own clinical research. Third, we did not conduct any statistical comparison between included studies. Therefore, we cannot exclude the possibility of selection bias, which may lead to biased conclusions and potentially overestimation of the plausible impact of sex on the response to PPI therapy.

## 9. Summary and Future Directions

Biological sex and sociocultural gender both seem to contribute to differences in PPI treatment indication, response, and potential outcomes. Secondary gastrin elevation seems to be a dose-dependent response to acid inhibition by PPI therapy, but there are also individual effects underlying the intra- and inter-individual variations in the observed serum gastrin levels following PPI treatment. The role of sex in gastrin elevation has not received much attention, and the inclusion of sex as a variable in studies on gastrin levels in patients on PPIs is often lacking. However, when sex-disaggregated data are presented, it seems that the female sex is a risk factor for gastrin elevation following PPI therapy. Further studies are needed to increase the understanding of the sex difference in gastrin elevation induced by PPIs, and it is still unclear whether this is a clinically important difference.

There is a constantly growing concern about the worldwide overuse of PPIs, especially their long-term use and potential adverse events. According to recent studies, females currently comprise more than half of PPI users and are involved in more ADRs in general. It is still unknown whether females are more sensitive to the inhibitory effect of PPIs, but gastrin is believed to play a role in the RAHS phenomenon after PPI withdrawal, which might maintain the need for further PPI use. The long-term consequences of continuous gastrin elevation remain unknown, but efforts should be made to examine the role of gastrin elevation in the overutilization of PPIs. Gastrin also exerts trophic effects on the GI tract mucosa; therefore, females might be at increased risk of developing morphological changes, such as FGPs in the gastric mucosa.

Gender and sex differences in PPI indications and sensitivity might have diagnostic and therapeutic influences. PPIs are now used for a variety of indications in a highly heterogeneous human population, making it difficult to generalize the findings from one study to another. Many studies are also aimed at symptomatology, and to a lesser extent, at physiology, thus putting little emphasis on the functions of PPIs in gastric acid inhibition and secondary gastrin elevation. However, it is conceivable that females might be at risk of PPI overuse, not only because they often receive higher milligram-per-kilogram doses, but they are also more likely than males to have burdens from non-acid-related symptoms. Such information is important to elucidate in the future. If the majority of female patients on long-term PPI therapy are able to step down or step off their PPI use, this could lead to major economic benefits for healthcare systems in many countries and potentially decrease the risks of side effects associated with long-term PPI use.

## 10. Conclusions

This review examined the literature related to the impact of female sex on the response to PPI therapy. Recent studies indicate that females on PPIs have increased gastrin release, which might contribute to differences in PPI treatment indication, response, and potential outcomes. Given the extensive use of PPI therapy, it seems prudent to address the potential issue of PPI overmedication of females, who could potentially benefit from receiving lower dosages of PPIs than those administered to males. The cornerstone of pharmacological therapy is to use the lowest effective dose to treat symptoms, and this is of course true for PPI therapy. Further studies are needed to increase the understanding of the sex difference in gastrin elevation induced by PPIs, and it is still unclear whether this is a clinically important difference. Also, the inclusion of sex- and gender-based analyses of PPI indication, response, and outcomes should be emphasized in future research.

## Figures and Tables

**Figure 1 pharmaceuticals-16-01722-f001:**
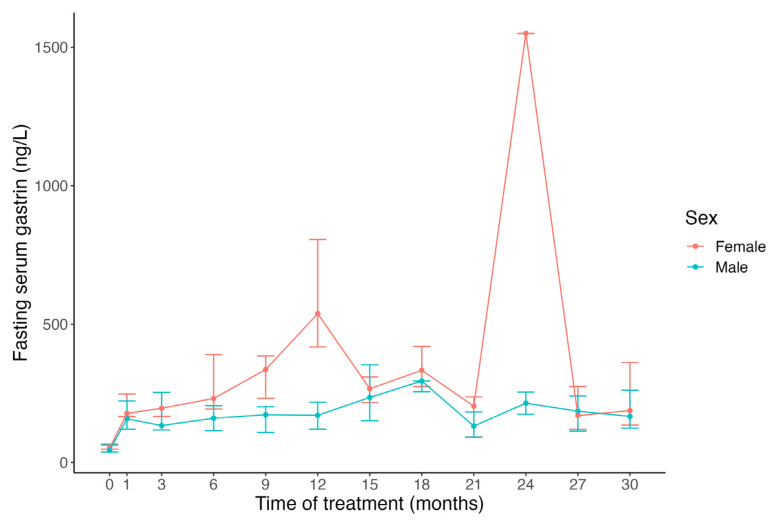
Median fasting serum gastrin levels in reflux esophagitis patients before treatment (0 months) and during omeprazole maintenance treatment. The graph is based on data from non-antrectomized patients (*n* = 29 (14 females)) [22]. Gastrin levels were higher in females (red line) than males (blue line). The error bars show the 25th and the 75th interquartile range [22]. Reprinted/adapted with permission from Ref. [22].

**Figure 2 pharmaceuticals-16-01722-f002:**
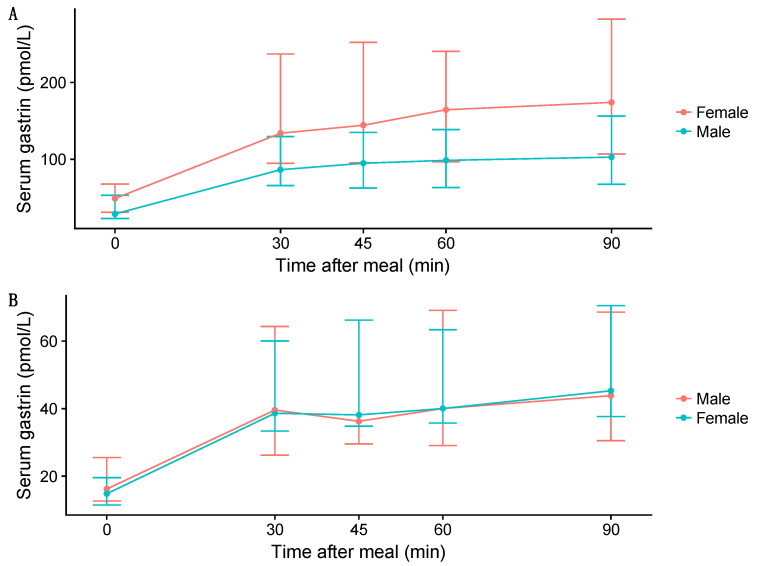
Median fasting (time = 0) and postprandial serum gastrin levels in patients with gastroesophageal reflux disease on long-term (over 2 years) proton pump inhibitor therapy (**A**) and controls not taking proton pump inhibitors (**B**). The graph is based on data from 100 patients (44 females) and 50 healthy controls (25 females) [28,29]. The error bars show the 25th and the 75th interquartile range. Female patients (red line) had significantly higher gastrin levels than males (blue line) pre- and postprandial, whereas such differences were not found in the control group (males in red, females in blue) [28]. Reprinted/adapted with permission from Ref. [28].

**Figure 3 pharmaceuticals-16-01722-f003:**
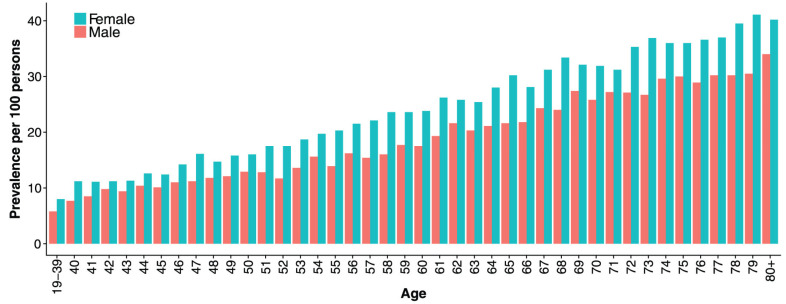
Sex- and sage-specific prevalence of proton pump inhibitor use among adults in Iceland in 2015. Reprinted with permission from Ref. [62]. 2018, copyright SAGE Publications.

**Table 1 pharmaceuticals-16-01722-t001:** Overview of studies that examined sex differences in gastrin release and/or gastrin elevation in healthy volunteers or patients receiving or not receiving PPI therapy.

Author, Year	N (M/F)	Subjects	Sex Difference	Baseline Gastrin (M vs. F)	Comments
Gedde-Dahl, 1974 [33]	298 (180/118)	Patients	No	67 (45) vs. 65 (40) (mean (SD))	Patients with different diseases undergoing pentagastrin test
Archimandritis et al., 1979 [34]	80 (43/37)	Healthy volunteers	No	54 (3) vs. 56.5 (3) pg/cc (mean (SEM))	No gender difference found 10 and 40 min after a meal
Feldman et al., 1983 [35]	41 (26/15)	Healthy volunteers	Yes	-	Females had higher basal and meal-stimulated gastrin levels, with average rises of 19 (2) vs. 53 (10) pg/mL (*p* < 0.001).
Prewett et al., 1991 [32]	131 (96/35)	Healthy volunteers	Yes	185 vs. 407	Gastrin values are 24 h integrated plasma gastrin concentrations (pmol⋅h⋅L^−1^).
Mossi et al., 1993 [36]	62 (30/32)	Healthy volunteers, *H. pylori* (-)	No	73 (5) vs. 74 (5) pg/mL (mean (SEM))	No association between sex or age and baseline gastrin levels
Jansen et al., 1990 [22]	32 (18/14)	EE patients	No	Baseline (*p* = NS)	At all-time intervals, females had higher fasting gastrin levels than males. Eight patients reached gastrin levels > 6 times the upper limit of normal range during follow-up (5 females).
		on PPIs	Yes	18 months (*p* < 0.01)	
			Yes	21 months (*p* < 0.05)	
Wang et al., 2010 [26]	95 (67/28)	BE and GERD patients on chronic PPIs	No	65 vs. 80 pM (mean) 40 vs. 47 pM (median)	No association between sex or age and baseline gastrin levels
Camilo et al., 2015 [37]	81 (13/68)	Chronic PPI users	-	-	Females were the only patients with gastrin levels > 115 pg/mL.
Shiotani et al., 2018 [20]	199 (143/56)	CV patients on PPIs, prophylaxis with aspirin	Yes	214 vs. 357 pg/mL	The F gender was associated with hypergastrinemia in a multiple logistic regression analysis, adjusted also for PPI use (vs. H2RAs and controls) and corpus atrophy.
Helgadottir et al., 2020 [21]	157 (79/78)	GERD patients on long-term PPIs	Yes	60 (42–90) vs. 92 (53–118) pg/mL (median (IQR))	Gastrin elevation was significantly associated with the F sex and PPI dosage.
Helgadottir et al., 2021 [38]	29 (14/15)	Healthy volunteers on short-term PPIs	Yes	Day 0 (12 vs. 7 pM)	Females had significantly higher baseline gastrin levels than males, but there was no significant difference between the sexes at the end of treatment (day 5).
			No	Day 5 (15 vs. 15 pM)	

Acronyms and abbreviations: BE = Barrett’s esophagus, CV = cardiovascular, EE = erosive esophagitis, F = female, GERD = gastroesophageal reflux disease, H2RAs = histamine type-2 receptor antagonists, IQR = interquartile range, M = male, NS = non-significant, PPI = proton pump inhibitor, SD = standard deviation, SEM = standard error of the mean.

**Table 2 pharmaceuticals-16-01722-t002:** Possible factors that could contribute to gender-related differences in gastrin release in healthy subjects and gastrin elevation secondary to PPI therapy.

Females have a lower number of parietal cells [39].
Females are less sensitive to gastrin effects on parietal cells [35].
Smaller stomachs of females, with more postprandial distension [40]
Gender differences in dietary intakes, lower energy density in females than in males [41]
Gender differences in gastric emptying, with slower gastric emptying in females [42,43]
Gender difference in body mass index (BMI) [44]
Sex hormones (unlikely, as no fluctuations in gastrin release throughout one menstrual cycle were observed in six females) [35]
Difference in metabolism of PPIs [45,46]

**Table 3 pharmaceuticals-16-01722-t003:** Overview of previous studies that have examined PPI deprescribing.

Author, Year	N (M/F)	Subjects	Type of Study	Sex Variable Mentioned	Type of PPI Deprescribing	Predictors and Other Comments
Inadomi et al., 2003 [60]	117 (112/5)	Patients with heartburn or acid regurgitatio	Prospective study	No	Step-down from multiple- to single-dose	Only the duration of PPI use before study predicted successful step-down (OR 0.66).
Bjornsson et al., 2006 [58]	96 (44/52)	Patients without history of PUD or EE	Double-blind, placebo-controlled trial	No	Step-off	GERD as PPI indication (OR 8.050) and serum gastrin (OR 1.018) predicted the need for reinstitution of PPIs after discontinuation.
Cote et al., 2007 [61]	223	GERD patients	Retrospective study	Very few females (~1%)	Step-down from BID to SID	Dose reduction was more successful in those without EE.
Fass et al., 2012 [13]	142 (62/80)	Symptomatic GERD patients	Single-blind trial	Yes	Step-down from BID to SID modified release PPI	No predictor was significant (age, sex, BMI, and baseline symptom scores). But OR for females not remaining well controlled was 0.499, NS.
Helgadottir et al., 2017 [12]	100 (51/49)	EE patients	Double-blind randomized trial	Yes	50% dose reduction	Successful step-down was predicted only by female sex with OR 1.3 (*p* = 0.048). Baseline fasting s-gastrin was NS (*p* = 0.49).
Hendricks et al., 2021 [59]	33 (19/15)	Patients with a clinical diagnosis of GERD	Randomized open-label trial	Yes	Step-off	Sex was not associated with resuming PPIs. H2RA use was associated with successful discontinuation of PPIs with HR 0.21 (*p* = 0.002).

Acronyms and abbreviations: BID = bis in die/twice a day, BMI = body mass index, EE = erosive esophagitis, F = female, GERD = gastroesophageal reflux disease, H2RAs = histamine type-2 receptor antagonists, HR = hazard ratio, M = male, NS = non-significant, OR = odds ratio, PPI = proton pump inhibitor, PUD = peptic ulcer disease, SID = semel in die/once a day.

**Table 4 pharmaceuticals-16-01722-t004:** The potential role of sex and gender on PPI overuse or overmedication.

Sex	Diagnosis	PPI Overmedication?
Female > Male	Non-erosive reflux disease Heartburn or regurgitation Extra-esophageal symptoms Comorbid anxiety or depression	Females at risk for overmedication because of partial symptom response and unnecessary PPI dose escalation for symptoms that are not related to acid reflux
Male > Female	Reflux esophagitis Barrett’s esophagus Esophageal adenocarcinoma	Females at risk for overmedication because they might be more sensitive to PPIs than males and could remain symptom control on lower PPI doses

## Data Availability

Data sharing is not applicable.

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
