# Peer review of "The Impact of Sex on the Response to Proton Pump Inhibitor Treatment"

_pharmaceuticals, 2023, doi:10.3390/ph16121722_

Round 1

Reviewer 1 Report

Comments and Suggestions for Authors

The article is well written and well organized. Topic is novel and interesting. Efforts for authors are appreciated. However, many comments should be addressed before final publication

1-     In the abstract, Add space between medicaltreatment

2-     The main side effects for hypergastrinemia should be illustrated with details in the abstract

3-     Add biological sex for keywords

4-     The following articles should be cited in the introduction

 Gender differences in symptoms in partial responders to proton pump inhibitors for gastro-oesophageal reflux disease. https://doi.org/10.1177/2050640614558343

Factors affecting response to proton pump inhibitor therapy in patients with gastroesophageal reflux disease: a multicenter prospective observational study

https://doi.org/10.1007/s00535-015-1073-0

Genetics of Response to Proton Pump Inhibitor Therapy

https://doi.org/10.2165/00129785-200303050-00002

5-     Lines 36-37 needs appropriate reference

6-     Generally, more diagrams should be included e.g. for the adverse effects of hypergastrinemia, main side effects that are prominent in female

7-     The reference numbering needs correction in table 1. For example Gedde-Dahl, 1974 (68) should be (26). Following normal numbering is strongly recommended.

8-     The following statements should be included in the abstract

[  A review of 12 pharmacokinetic studies suggested a sex difference in two pharmaco- kinetic parameters. The area under the curve (AUC) and maximum concentrations  (Cmax) values were approximately 30% higher in females than in males after a single dose, 124 with less difference during repeated administration.]

[compared with males, females  might be more sensitive to PPIs’ inhibitory effects on acid secretion and induced gastrin  release.]

9-     Table 2 should be represented in colured diagram

10-  In line 161, add full name for GERD with short description for GERD symptoms

11-  Short definition of  fundic gland polyps (FGPs) is required in line 247

12-  General recommendations based on findings in this review should be summarized in a brief paragraph

Best wishes

Reviewer 2 Report

Comments and Suggestions for Authors

The current manuscript, about the impact of gender on the response to proton pump inhibitor treatment has some interesting aspects, including the tentative association of gender with several relevant factors regarding these medications. Nevertheless, overall the evidence seems to be quite scarce. Many times, the conclusion is that there is no conclusion. Additionally, improvements are needed, including:

- The abstract should include a part regarding the discussion and conclusion of the review;

- How were the analyzed studies selected? The criteria should be evident an mentioned; were there any relevant studies excluded from analysis? If so, why? What were the used databases, keywords, etc.?;

- A schematic representation (figure) regarding the factors surrounding the effect of gender in PPI should be made and added;

- The direct comparison between included studies should be better done; additionally, whenever possible a statistical analysis should be added to better support the comparison;

- The article is quite short; the discussion should be improved, regarding each individual study, but also the comparison between the studies;

- A table should be added on each section to better summarize the results of the included studies;

- Images from the included studies should be added, with adequate permission from the original journal, for better reader visualization.

Round 2

Reviewer 1 Report

Comments and Suggestions for Authors

the authors did most of the recommendations. the paper could be published in the current form. 

Greetings 

Author Response

We are very thankful

Reviewer 2 Report

Comments and Suggestions for Authors

Although the authors have performed some changes that have improved the manuscript, it is still not sufficient:

- From the previous comment “A schematic representation (figure) regarding the factors surrounding the effect of gender in PPI should be made and added” and authors’ response: “The point is well taken. Graphical abstract has been added which is enclosed.”, the point was not to add a graphical abstract, but figures embedded within the text, since the graphical abstract is already a mandatory requirement and should exist by itself;

- From the previous comment “The article is quite short; the discussion should be improved, regarding each individual study, but also the comparison between the studies” and authors’ response: “We respectfully disagree with the reviewer and we do not think that the paper would be better if it would much longer than it is. By providing more systematic comparisons with other studies in the new tables, we think this goal has been achieved.”, in my opinion this is not adequate, since there are several sections that lack improvement and better discussion, such as “7. Other side effects of PPIs” (it is hardly likely that there is nothing more to say about side effects, more should be discussed, including molecular mechanisms of how they might occur);

- From the previous comment “Images from the included studies should be added, with adequate permission from the original journal, for better reader visualization.” and authors’ response: “We agree that an increased number of images can provide better visualization, Thus, we have added a graphical abstract containing 6 figures, with mostly results from previous studies.”, again, this is not to be done through the graphical abstract, but by adding the images in the main body of the manuscript.

- The limitations of this review should be added, including the fact that the analyzed studies have important differences between them (such as number of volunteers, healthy vs ill condition), which makes it hard to draw solid generalized conclusions.

Author Response

We thank the reviewer again for the comments, we have made new changes that are highlighted in green. Here are our point by point responses:

Although the authors have performed some changes that have improved the manuscript, it is still not sufficient:

  1. From the previous comment “A schematic representation (figure) regarding the factors surrounding the effect of gender in PPI should be made and added” and authors’ response: “The point is well taken. Graphical abstract has been added which is enclosed.”, the point was not to add a graphical abstract, but figures embedded within the text, since the graphical abstract is already a mandatory requirement and should exist by itself;

Response: As suggested by the reviewer it might be improve the paper to add figures. Three figures have been embedded in the text now: 

Figure 1 is adapted from data published in table 1 from https://pubmed.ncbi.nlm.nih.gov/2199288/. And this article is added as reference in the figure text.

Figure 2 is adapted from data published in https://pubmed.ncbi.nlm.nih.gov/24210828/. Not the same figure, but figure from first author PhD essay and both the article and PhD is added as reference in the figure text.

Figure 3 is from  https://pubmed.ncbi.nlm.nih.gov/29872455/. We have permission (license no. 5682450985030) for re-use of the figure. This is stated in the figure text.

  1. From the previous comment “The article is quite short; the discussion should be improved, regarding each individual study, but also the comparison between the studies” and authors’ response: “We respectfully disagree with the reviewer and we do not think that the paper would be better if it would much longer than it is. By providing more systematic comparisons with other studies in the new tables, we think this goal has been achieved.”, in my opinion this is not adequate, since there are several sections that lack improvement and better discussion, such as “7. Other side effects of PPIs” (it is hardly likely that there is nothing more to say about side effects, more should be discussed, including molecular mechanisms of how they might occur);

Response: We agree with the reviewer that something more can be said about PPIs, but the current paper is a part of a series of papers within the journal on gender related issues in pharmacology. However, we have added text accordingly in chapter 7, which has in our opinion been improved. We have also added type of study in table 3 for better comparison between the studies.

  1. From the previous comment “Images from the included studies should be added, with adequate permission from the original journal, for better reader visualization.” and authors’ response: “We agree that an increased number of images can provide better visualization, Thus, we have added a graphical abstract containing 6 figures, with mostly results from previous studies.”, again, this is not to be done through the graphical abstract, but by adding the images in the main body of the manuscript.

Response: See response to point 1.

  1. The limitations of this review should be added, including the fact that the analyzed studies have important differences between them (such as number of volunteers, healthy vs ill condition), which makes it hard to draw solid generalized conclusions.

Response: The point is well taken and chapter on limitations has been added.

Round 3

Reviewer 2 Report

Comments and Suggestions for Authors

Major issues have been addressed.